# Provision of Energy- and Wavelength-Efficient Traffic Grooming for Sparse WDM-Enabled Distributed Satellite Cluster Networks

**Cong Peng [1,2], Yuanzhi He [1,\*], Di Yan [1], Huajun Fu [3] and Shanghong Zhao [2]**

1   Systems Engineering Research Institute, Academy of Military Sciences, Beijing 100141, China; 17791760275@163.com (C.P.); yandi_2022@126.com (D.Y.)
2   The Communication System Teaching and Research Section, Information and Navigation College, Air Force Engineering University, Xi'an 710077, China; zhaoshangh@aliyun.com
3   Systems Science and Engineering College, Sun Yat-sen University, Guangzhou 511436, China; fuhj@mail2.sysu.edu.cn
\*   Correspondence: he_yuanzhi@126.com

**Abstract:** Sparse wavelength division multiplex (WDM) enabled distributed satellite cluster networks (DSCNs) have emerged as a promising architecture to accommodate future extensive applications. Networking of the DSCNs will face the challenges of explosively increasing traffic requests, the limited number of wavelengths, and restricted energy provisioning. To address these issues, a novel approach, the two-phase traffic grooming based on the matching algorithm (TPTG_MA), is proposed in this paper. To analyze resource utilization, energy- and wavelength- minimized models are established. After that, we develop the MA to tackle the traffic grooming problem in two phases, including the first phase for traffic aggregation and sub-wavelength assignment (TAASA) and the second phase for sub-wavelength grooming (SG). To evaluate the performance of the proposed TPTG_MA, the direct lightpath grooming (DLG) heuristic and the genetic algorithm (GA) are simulated for comparison. The results demonstrate that the TPTG_MA and DLG_GA outperform TPTG and DLG in the average wavelength utilization ratio (AWUR), the energy consumption saving (ESC), and the blocking probability. Compared with the DLG_GA, the TPTG_MA achieves at most 18% and 23% higher AWUR in the 12-node and 22-node topologies, respectively. In addition, the TPTG_MA can actualize at most 10% ECS improvement over the DLG_GA. At last, the influence of the network size, the number of wavelengths, and the number of hops are discussed.

**Keywords:** distributed satellite cluster networks; sparse WDM; traffic grooming; matching algorithm

## 1. Introduction

Due to the extension of terrestrial 5G to satellite networks, the off-the-shelf microwave communication can no longer address the requirements of future extensive applications, such as remote sensing, Internet of Vehicles (IoV), and smart ocean [1–3]. Free-space optical communication has the superiorities of large-capacity, low-latency, excellent anti-interference, and terminal miniaturization, offering a promising means for satellite networking [4,5]. Leveraging the WDM technique, the capacity of inter-satellite links (ISLs) can be greatly heightened to accommodate various services [6]. However, unlike the dense WDM-enabled terrestrial fiber links, wavelengths in each ISL are limited. On the one hand, the limited bandwidth of onboard optical amplifiers decides the sparse WDM ISLs. On the other hand, the crosstalk of filtering devices decreases the number of wavelengths in each ISL [7].

Driven by the limited single-satellite payload and massive data traffic requests, the distributed satellite cluster (DSC) has been designed to construct a large platform from the system perspective over the decade [8–12]. Satellites in the DSC are arranged on the same

or adjacent orbits to orderly and jointly accomplish spatial tasks. The typical architecture includes the F6 (Future, Fast, Flexible, Fractionated, Free-Flying) Program, space-based group, and GRACE [10–12]. The DSCNs are constructed by connecting multiple DSCs. With the aid of high-speed laser ISLs, multigranular information, such as broadband remote sensing information and the narrowband Internet of Things (IoT) information, can be quickly transmitted and distributed in the DSCNs. Although the overall ability of the DSC system is enhanced via multi-satellite collaboration and WDM ISLs, energy consumption is a critical issue that cannot be neglected. Because the satellite is a kind of energy harvest system, the harvested energy is used for orbit keeping, signal processing, and ISLs establishment and maintenance [13].

A promising solution for wavelength and energy savings in the sparse WDM-enabled DSCNs is utilizing traffic grooming, where multiple traffic requests are multiplexed into a single lightpath that is then transmitted at the optical layer. Traffic grooming has been widely investigated in terrestrial optical networks over the past decades [14–18]. However, to our knowledge, no previous work has been focused on traffic grooming in optical satellite networks (OSNs) and considering energy and wavelength efficiency as well. In addition, most existing heuristics proposed for traffic grooming belong to DLG where traffic requests are directly groomed into wavelengths [16–18]. To actualize traffic grooming, the hybrid cross-connect architecture is maintained in every satellite node, including the optical cross-connect (OXC) unit which is functioned as wavelength routing operations, and the digital cross-connect (DXC) unit which is applied for electric switching and flow aggregation [16]. In this paper, we introduce the sub-wavelength, (e.g., OC-1) as the intermediate granularity and divide the traffic grooming into two phases, i.e., the TAASA and the SG. On the one hand, we can achieve a further segmentation of the fixed-capacity wavelength, targeting to enhance the utilization efficiency of wavelengths. On the other hand, the group transmission of traffic requests is realized, which helps to save transmission resources, especially for energy and wavelength channels. To obtain an optimal grooming scheme, the traffic requests, sub-wavelengths, and wavelengths can be treated as three sets of players to be matched with each other to minimize the energy consumption and the used wavelengths. We then resort to the matching algorithm to resolve the traffic grooming problem.

The rest of the paper is structured as follows. The scenario of traffic grooming in the DSCNs is described, and the energy- and wavelength- minimized models are established in Section 2. Section 3 designs the matching algorithm to solve the problem in two phases. The performance evaluation and analysis of the proposed algorithm are presented in Section 4. The conclusions are drawn in Section 5.

## 2. Scenario and Models of Traffic Grooming in the DSCNs

As shown in Figure 1, the DSCNs including a group of U satellites provide service for users within its coverage. Through optical ISLs, large-capacity optical networks can be constructed to support long-distance transmission for large regions [19–21]. In this paper, we neglect the impact of the specific terrestrial users and assume that satellites in the DSCNs serve as the source and destination nodes. It is equivalent to the result that satellites have collected the traffic requests of users within their coverage. As depicted in Figure 1, data flow 1, 2, and 3 originate from satellites 2, U, and 3, and then terminate at satellites 3, 2, and 4, respectively.

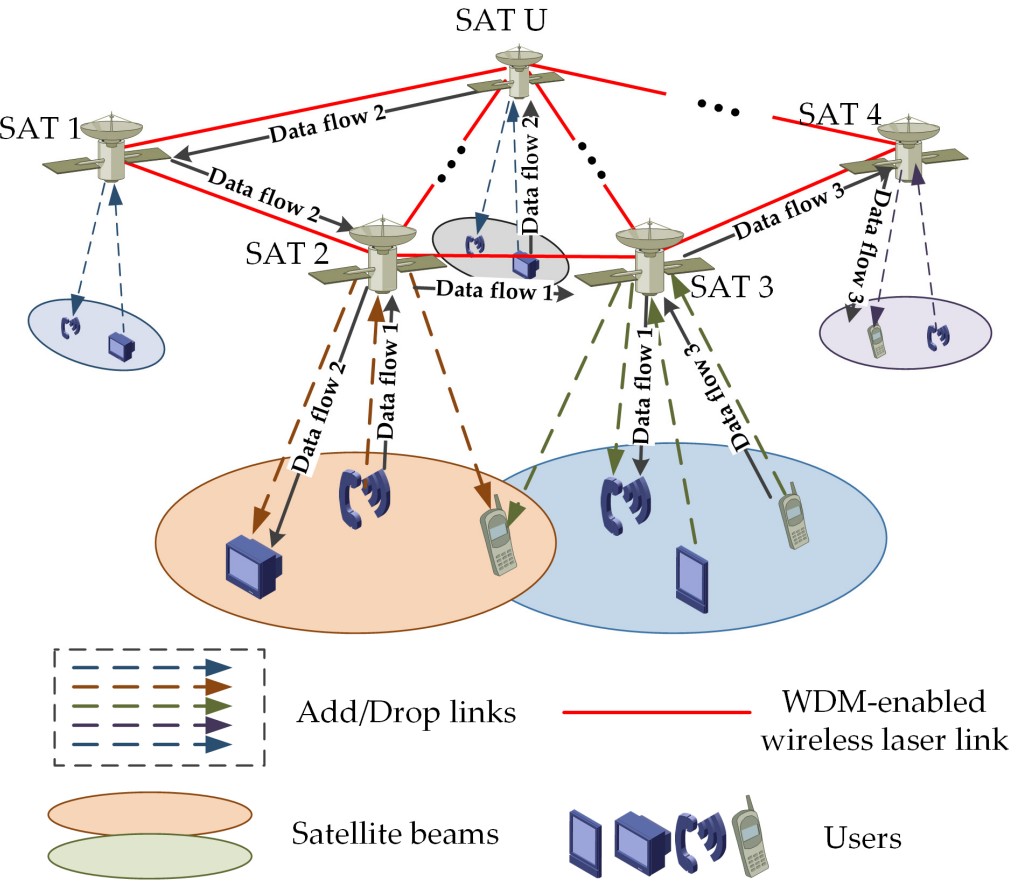

**Figure 1.** A group of U satellites is interconnected by sparse WDM-enabled wireless laser links, providing services for users within its coverage, where three data flows traverse from the sources to destinations.

### 2.1. Scenario Description

To clearly depict the DSCNs, a bidirectional graph $G(V, E)$ is established to describe the physical topology, where $V$ ($V = \{V_1, V_2, \cdots, V_U\}$) denotes the satellite nodes and $E$ represents the ISLs. Equipped with multiple independent antenna units, the single satellite can communicate with its neighboring nodes using sparse WDM-enabled large-capacity ISLs. It is assumed that each ISL can modulate up to $W$ wavelengths and the optical antenna unit is tunable to guarantee the continuity of wavelengths. Traffic requests in the DSCNs exhibit the characteristic of multiple granularities, and the bandwidth demand matrix can be expressed as $\mathcal{B} \in \mathbb{R}^{U \times U}$, where each vector represents the bandwidth demand of a traffic request, i.e., $\mathcal{B}[s, d] = \left\{ b_{s,d}^f | s, d \in V, 1 \leq f \leq F_{s,d} \right\}$ and $F_{s,d}$ is the number of traffic requests from $s$ to $d$.

Under given traffic demands, there are two objectives to be optimized, i.e., the minimum energy consumption and the minimum number of wavelengths. To calculate the energy consumption, we use $E_{oe}$, $E_{eo}$, $E_{agg}$, and $E_{edfa}$ to represent the energy consumption of each port for optical/electric conversion, electric/optical conversion, traffic aggregation, and EDFA amplifier, respectively. The transmitting energy can be expressed as $E_{TX}$. Other notations are given as follows.

### 2.1.1. Basic Node Pair Representation

| | |
|---|---|
| $(m, n)$ | Originating and terminating satellite nodes in the physical topology $G$, which also serves as the endpoints of an ISL. |
| $(i, j)$ | Originating and terminating nodes of a virtual lightpath, which traverses through several ISLs in the physical topology. Each virtual lightpath is allocated a wavelength. |
| $(s, d)$ | Source and destination nodes of an end-to-end connection request, or aggregated request, which will be routed on the virtual lightpaths. |

### 2.1.2. Sets and Parameters

| | |
|---|---|
| $Q_m$ | Set of visible satellites for satellite $m$. |
| $\alpha_m$ | The node degree of satellite $m$; $2 \leq \alpha_m \leq |Q_m|$ and $\alpha_m \in N^+$. This implies that each satellite in the DSCNs should be connected by at least two and at most $|Q_m|$ ISLs. |
| $\Omega$ | Capacity of a wavelength and $\Omega = \delta \, \Omega_s$, where $\Omega_s$ is the capacity of a sub-wavelength and $\delta$ is the number of sub-wavelength-level aggregated flows that a wavelength can carry. |
| $\Delta_m^{agg}$ | Number of ports used for aggregating the low-speed traffic requests in satellite $m$. |
| $\Delta_m^{oe}$ | Number of ports used for optical/electric conversion in satellite $m$. |
| $\nabla_m^o$ | Total number of optical ports in satellite $m$. |
| $\nabla_m^e$ | Total number of electric ports in satellite $m$. |
| $I_{mn}$ | Number of ISLs from $m$ to $n$; If $m \in Q_n$ or $n \in Q_m$, $I_{mn} = 1$, otherwise, $I_{mn} = 0$. |
| $L_{ij}$ | Number of virtual lightpaths from $i$ to $j$; $L_{ij} = L_{ij}^w$, where $L_{ij}^w$ is the number of lightpaths from $i$ to $j$ on wavelength $w$. |
| $\Gamma_{ij}^{sd}$ | Number of connection requests between node pair $(s, d)$ which passes through virtual lightpath $(i, j)$. |
| $\Theta_{mn}^{ij}$ | Number of virtual lightpaths from $i$ to $j$ that traverse ISL $(m, n)$. |
| $\Psi_{ij}^f$ | Real capacity of lightpath $(i, j)$ which is occupied by the $f$-th connection request and $\Psi_{ij}^f \leq \Omega_s, \forall 1 \leq f \leq F_{s,d}, \, i, j \in V$. |

### 2.1.3. Decision Variables

| | |
|---|---|
| $\mu_{ij}^w$ | Binary variable related to wavelength assignment, which equals 1 if there is a lightpath from $i$ to $j$ on wavelength $w$; otherwise, 0; $\Theta_{mn}^{ij} = \sum_w \mu_{ij}^w, \forall m, n$. |
| $\xi_{f,p}$ | Binary variable equals to 1 if the $f$-th connection request is aggregated into the $p$-th sub-wavelength; otherwise, 0. |
| $\vartheta_{p,w}$ | Binary variable equals to 1 if the $p$-th sub-wavelength is groomed onto wavelength $w$; otherwise, 0; $\delta = \sum_p \vartheta_{p,w}, \forall w$. |
| $Z_{ij}^{f,p,w}$ | The value is 1 when the $f$-th connection request is aggregated into the $p$-th sub-wavelength which traverses lightpath $(i, j)$ on wavelength $w$; otherwise, 0. |
| $R_{mn}^{ij,w}$ | The value is 1 when the lightpath $(i, j)$ traverses ISL $(m, n)$ on wavelength $w$; otherwise, 0. |

### 2.2. Optimization Models

### 2.2.1. Minimize the Energy Consumption

$$\min E^{Total} = \sum_k E_k^{total}$$
$$= \sum_k \left( E_k^{sou} + E_k^{des} + E_k^{int} + \sum_{n \in Q_k} I_{kn} E_{TX} \right) \tag{1}$$

where

$$
\begin{aligned}
E_k^{sou} = \sum_{d:d\neq k} &\left[ \left\lceil \frac{\sum_{f\in[1,F_{k,d}]} b_{k,d}^f}{\Omega_s} \right\rceil \cdot \left( E_{agg} + \frac{E_{eo}+E_{edfa}}{\delta} \right) \right. \\
&\left. \cdot \min(1, \sum_p \xi_{f,p} \cdot \sum_j \sum_{i\neq j} Z_{ij}^{f,p,w}) \right], \ \forall k\in V, w\in[1,W]
\end{aligned}
$$

$$
\begin{aligned}
E_k^{des} = \sum_{s:s\neq k} &\left[ \left\lceil \frac{\sum_{f\in[1,F_{s,k}]} b_{s,k}^f}{\Omega_s} \right\rceil \cdot \left( E_{agg} + \frac{E_{oe}}{\delta} \right) \right. \\
&\left. \cdot \min(1, \sum_p \xi_{f,p} \cdot \sum_j \sum_{i\neq j} Z_{ij}^{f,p,w}) \right], \ \forall k\in V, w\in[1,W]
\end{aligned}
\tag{2}
$$

$$
E_k^{int} = \sum_{s\neq k}\sum_{d\neq k}\sum_{j\neq k}\sum_{i\neq k} \left( \sum_{n:n\in Q_k} \Gamma_{ij}^{sd}\Theta_{kn}^{ij} + \sum_{m:m\in Q_k} \Gamma_{ij}^{sd}\Theta_{mk}^{ij} \right) \cdot E_{edfa}
$$

The minimum energy consumption of satellite $k$ can be calculated by $E_k^{Total}$ in Equation (1), where $E_k^{sou}$, $E_k^{des}$, and $E_k^{int}$ are the used energy when $k$ serves as the source, destination, and intermediate nodes, respectively. The term $I_{kn}E_{TX}$ in (1) represents the energy consumed by ISL establishing and maintaining between satellite $k$ and $n$, (i.e., the energy consumption of the transceivers). The term $\left\lceil \sum_{f\in[1,F_{k,d}]} b_{k,d}^f/\Omega_s \right\rceil$ in (2) is the number of aggregated connection requests, which originate from $k$ and terminate at $d$. It is assumed the number of ports used for electric/optical conversion is equivalent to that for the EDFA amplifier. The term $\min(1, \sum_p \xi_{f,p} \cdot \sum_j \sum_{i\neq j} Z_{ij}^{f,p,w})$ indicates that the aggregated traffic requests are taken as a whole in the process of optical/electric conversion and optical amplification. If the satellite $k$ serves as the intermediate node, it is responsible for the signal receiving and forwarding, and the number of corresponding occupied ports can be expressed as $\sum_{m:m\in Q_k} \Gamma_{ij}^{sd}\Theta_{mk}^{ij}$ and $\sum_{n:n\in Q_k} \Gamma_{ij}^{sd}\Theta_{kn}^{ij}$, respectively.

Moreover, many constraints must be satisfied in the optimization model. First, we list the port constraints $C_1\sim C_3$, where $C_1$ means that the occupied ports cannot exceed the maximum number of ports for traffic aggregation, $C_2$ represents that the optical/electric and electric/optical conversions of traffic requests must be completed within the limited $\Delta_k^{oe}$. Apart from ports used for traffic transformation, other ports are responsible for lightpath bypass, which is depicted as $C_3$.

$$
\begin{aligned}
C_1 &: \sum_{d:d\neq k} \left\lceil \sum_{f\in[1,F_{k,d}]} b_{k,d}^f/\Omega_s \right\rceil \leq \Delta_k^{agg} = \nabla_k^e - \delta\,\Delta_k^{oe} \\
C_2 &: \frac{1}{\delta}\left( \sum_{d:d\neq k} \left\lceil \sum_{f\in[1,F_{k,d}]} b_{k,d}^f/\Omega_s \right\rceil + \sum_{s:s\neq k} \left\lceil \sum_{f\in[1,F_{s,k}]} b_{s,k}^f/\Omega_s \right\rceil \right) \leq \Delta_k^{oe} \\
C_3 &: \sum_{n:n\in Q_k} \Gamma_{ij}^{sd}\Theta_{kn}^{ij} + \sum_{m:m\in Q_k} \Gamma_{ij}^{sd}\Theta_{mk}^{ij} \leq \nabla_k^o - \Delta_k^{oe}, \quad \forall s,d,i,j\in V
\end{aligned}
\tag{3}
$$

Second, we list other constraints $C_4\sim C_7$. Wavelength constraint $C_4$ is utilized to guarantee that the number of wavelengths used in every ISL cannot exceed $W$. It is assumed that lightpaths in an ISL comprise both the forward and the reverse ones. $C_5$ ensures that each traffic flow can traverse at most one virtual lightpath on any wavelengths, which implies that the traffic flows cannot be split in the process of forwarding. The flow conservation constraint is described as $C_6$, where connection requests originating from the same nodes terminate at different destinations via several virtual lightpaths. Capacity constraint is

provisioned in $C_7$ to assure that massive traffic flows can be aggregated as long as the maximum capacity of sub-wavelength is not exceeded.

$$
\begin{aligned}
C_4 &: \sum_i \sum_{j \neq i} (\Theta_{mn}^{ij} + \Theta_{nm}^{ij}) \leq W, \ \forall m, n \in V \\
C_5 &: \sum_p \sum_{w \in [1,W]} \vartheta_{p,w} Z_{ij}^{f,p,w} \leq 1, \ \forall f \in [1, F_{s,d}], i, j \in V \\
C_6 &: \sum_{j \in V: j \neq i} \Gamma_{ij}^{sd} - \sum_{j \in V: j \neq i} \Gamma_{ji}^{sd} = \begin{cases} F_{s,d}, \ i = s \\ -F_{s,d}, \ i = d \ , \ \forall s, d \in V \\ 0, \ otherwise \end{cases} \\
C_7 &: \sum_{f', f' \neq f} b_{s,d}^{f'} Z_{ij}^{f',p,w} \leq \Omega_s - \Psi_{ij}^f, \ \forall s, d, i, j, p, w
\end{aligned}
\tag{4}
$$

### 2.2.2. Minimize the Number of Wavelengths

$$
\min \sum_w \left( \sum_{i,j:i \neq j} \mu_{ij}^w + \sum_{i',j':i' \neq j', P_{i'j'} \cap P_{ij} = \varnothing} \mu_{i'j'}^w \right)
\tag{5}
$$

$$
s.t. \ \mu_{ij}^w = \prod_{m,n \in Q_m} R_{mn}^{ij,w} I_{mn}, \ \forall i, j \in V, w \in [1, W]
\tag{6}
$$

$$
\sum_w \left( \sum_{n \in Q_m} R_{mn}^{ij,w} - \sum_{n \in Q_m} R_{nm}^{ij,w} \right) = \begin{cases} L_{ij}, \ m = i \\ -L_{ij}, \ m = j \ , \ \forall i, j \in V \\ 0, \ otherwise \end{cases}
\tag{7}
$$

$$
\sum_w \left( \sum_j \sum_{i \neq j} Z_{ij}^{f,p,w} \sum_{m \in P_{ij}} \sum_{n \in P_{ij}, n \neq m} R_{mn}^{ij,w} \right) = \xi_{f,p}, \ \forall f, p
\tag{8}
$$

and $C_1 \sim C_5$, $C_7$.

The objective function aims to minimize the required wavelengths, which are calculated in (5). It is obvious that a wavelength can be exploited more than once, so long as the virtual paths do not coincide. Constraint (6) ensures that there is at most one wavelength traversing virtual lightpath $(i, j)$ at any ISL, where the binary variable $R_{mn}^{ij,w}$ is used to implement routing selection and wavelength assignment. Constraint (7) presents the flow conservation limitation in the optical layer, that is, the number of virtual lightpaths can be expressed as many segments of ISLs. Constraint (8) determines the state of sub-wavelength grooming, in other words, the matching results between the connection requests and the sub-wavelengths.

## 3. Traffic Grooming Algorithm Design

The parameters defined in the last section are included in constraints and objective functions, and once the values of these parameters are determined, the objective functions are only influenced by the defined decision variables. Consequently, we can resort to heuristic algorithms to solve the energy- and wavelength- minimized traffic grooming problem. To formulate the traffic grooming, we introduce the sub-wavelength as the intermediate granularity between the basic requests and wavelengths. Consequently, the traffic grooming problem can be formulated as a two-phase ILP problem, i.e., the first phase for TAASA and the second phase for SG. For analytical derivation, we regard the two-phase traffic grooming as a multivariate matching process. Specifically, the connection requests, sub-wavelengths, and wavelengths are three sets of players to be matched with each other to minimize the energy consumption and the used wavelengths, while the interdependencies exist among the requests due to the flow aggregation mechanism. It enables us to solve the two-phase ILP problem by utilizing the matching algorithm.

### 3.1. Traffic Aggregation and Sub-Wavelength Assignment Algorithm

(1) Definitions: The collection of all traffic requests is denoted by $\mathcal{S} = \{\mathcal{S}_1, \mathcal{S}_2, \cdots, \mathcal{S}_N\}$, and the corresponding path set can be expressed as $L_\mathcal{S} = \{L_{\mathcal{S}_1}, L_{\mathcal{S}_2}, \cdots, L_{\mathcal{S}_N}\}$. Noting that there is more than one path available for a request, thus we have $L_{\mathcal{S}_n} = \{l_1, l_2, l_3\}$, where the $l_1$, $l_2$, and $l_3$ represent the shortest, the second shortest, and the third shortest path, respectively. The requests can be divided into $M$ nonoverlapped groups, i.e., $\mathcal{U} = \{\mathcal{U}_1, \mathcal{U}_2, \cdots, \mathcal{U}_M\}$ where $\mathcal{U}_m = \{\mathcal{S}_{m,1}, \mathcal{S}_{m,2}, \cdots, \mathcal{S}_{m,N_m}\}$ and the corresponding paths satisfy $L_{\mathcal{S}_{m,1}} \subseteq L_{\mathcal{S}_{m,2}} \subseteq \cdots L_{\mathcal{S}_{m,N_m}}$. The requests within a group will be aggregated into a sub-wavelength. For any two groups $\mathcal{U}_i$ and $\mathcal{U}_j$, where $\mathcal{U}_i \cap \mathcal{U}_j = \varnothing$, the preference $\mathcal{U}_i \prec_{\mathcal{S}_n} \mathcal{U}_j$ indicates that the request $\mathcal{S}_n$ is willing to be a part of group $\mathcal{U}_j$, rather than group $\mathcal{U}_i$.

(2) Preference relation: Since the traffic transformation contributes to the major energy consumption, the preference of any request depends on the total used energy. Therefore, we denote the matching utility of the traffic aggregation as below.

$$\Xi(\mathcal{S}, \mathcal{U}) = -\sum_k \left( E_k^{sou} + E_k^{des} + E_k^{int} + \sum_{n \in Q_k} I_{kn} E_{TX} \right) \tag{9}$$

Our target is to minimize the total energy consumption meanwhile meeting the constraints $C_1 \sim C_7$. Therefore, the preference of the request improves when the matching utility increases. Requests can swap among different groups if less energy consumption can be actualized. The swap operation of any request is decided by the strict preference. By compare-and-swap operations, the preference of all requests will reach and keep a final equilibrium state where the minimum energy consumption is achieved. For any request $\mathcal{S}_n$ in group $\mathcal{U}_m$, the preference is defined as below.

$$\mathcal{U}_i \prec_{\mathcal{S}_n} \mathcal{U}_j \Leftrightarrow \Xi(\mathcal{U}_i) + \Xi(\mathcal{U}_j) > \Xi(\mathcal{U}_i \backslash \{\mathcal{S}_n\}) + \Xi(\mathcal{U}_j \cup \{\mathcal{S}_n\}) \tag{10}$$

where $\Xi(\mathcal{U}_i)$ is the matching utility of the traffic aggregation in group $\mathcal{U}_i$, which satisfies $\mathcal{U}_j \in \mathcal{U} \cup \varnothing$ and $\mathcal{U}_i \cap \mathcal{U}_j = \varnothing$. In this situation, the sum energy consumption of all connection requests in group $\mathcal{U}_i$ and $\mathcal{U}_j$ is strictly decreased if the request $\mathcal{S}_n$ moves from group $\mathcal{U}_i$ to group $\mathcal{U}_j$. If the sum energy consumption of all connection requests in the changed groups is increased, the request $\mathcal{S}_n$ tends to stay in the current group $\mathcal{U}_i$, as shown below.

$$\mathcal{U}_i \succ_{\mathcal{S}_n} \mathcal{U}_j \Leftrightarrow \Xi(\mathcal{U}_i) + \Xi(\mathcal{U}_j) \leq \Xi(\mathcal{U}_i \backslash \{\mathcal{S}_n\}) + \Xi(\mathcal{U}_j \cup \{\mathcal{S}_n\}) \tag{11}$$

(3) Algorithm design for TAASA

If the connection request $\mathcal{S}_n$ is proposed by more than one sub-wavelength, it will select the sub-wavelength $\mathcal{U}_i$ with the minimum sum energy consumption from the candidates and reject others. The objective of the matching algorithm is to search the matching pairs, and the key operation encompasses proposing phase and rejecting phase. In the scenario of traffic aggregation, each sub-wavelength is allowed to propose to more than one connection request so long as the minimum total energy consumption is guaranteed.

Based on the defined preference relation, each connection request can decide the swap operations for different groups. To quickly obtain the matching results, each unmatched sub-wavelength $\mathcal{U}_i$ proposes to an unmatched connection request $\mathcal{S}_n$ to match with, which satisfies

$$(\mathcal{S}^*, \mathcal{U}^*) = \arg \min_{\mathcal{S}_n \in \mathcal{S}, \mathcal{U}_i \in \mathcal{U} \cup \varnothing, \mathcal{S}_n \cap \mathcal{U}_i = \varnothing} \Xi(\mathcal{S}, \mathcal{U}) \tag{12}$$

The whole matching algorithm for traffic aggregation is presented in detail as Algorithm 1. At first, connection requests are allocated to different groups according to the path affiliation. Then, the proposing and rejecting operations are performed to aggregate all the

connection requests into different sub-wavelengths. The following compare-and-swap operations ensure that each connection request is aggregated into the target sub-wavelength. The iterations will not stop until no connection request is willing to be a part of other groups anymore.

---

**Algorithm 1:** TAASA algorithm

---

**Input:** Sets of connection requests $\mathcal{S}$ and its corresponding paths $L_{\mathcal{S}}$, sub-wavelengths (groups) $\mathcal{U}$, and the matching utility function $\Xi(\mathcal{S},\mathcal{U})$.
**Output:** Matching pairs between connection requests and sub-wavelength.
1. Initialization
2. Allocate all connection requests initially to groups according to the path affiliation.
3. **Proposing and rejecting**
4. **while** at least one connection request is unmatched **do**
5. 　Sub-wavelength $\mathcal{U}_m$ proposes to connection request $\mathcal{S}_n$ according to (12).
6. 　　**if** connection request $\mathcal{S}_n$ is proposed by more than one sub-wavelength **then**.
7. 　　Connection request $\mathcal{S}_n$ selects the sub-wavelength $\mathcal{U}_m$ with the minimum sum energy consumption $\Xi(\mathcal{S},\mathcal{U})$ from the candidates and rejects other proposals.
8. 　　**else**
9. 　　　Connection request $\mathcal{S}_n$ is matched with the proposing sub-wavelength.
10. 　　　Connection request $\mathcal{S}_n$ is removed from $\mathcal{S}$.
11. 　　**end if**
12. **end while**
13. **Compare-and-swap operations**
14. **repeat**
15. **for** connection request $n = 1 \rightarrow N$, where $\mathcal{S}_n \in \mathcal{S}$
16. 　**for** group $\mathcal{U}_m = \mathcal{U}_1 \rightarrow \mathcal{U}_M$, where $\mathcal{U}_k \cap \mathcal{U}_l = \varnothing$, $1 \leq k, l \leq M$
17. 　　Calculate the sum energy consumption for group $\mathcal{U}_k$ and $\mathcal{U}_l$.
18. 　　Connection request $\mathcal{S}_n$ moves from group $\mathcal{U}_k$ to group $\mathcal{U}_l$.
19. 　　Calculate the sum energy consumption for the updated group $\mathcal{U}_k$ and $\mathcal{U}_l$, respectively.
20. 　　Compare the change of the sum energy consumption.
21. 　　**if** the sum energy consumption decreases by the swap operation
22. 　　Connection request $n$ stays in group $\mathcal{U}_l$.
23. 　　**else** connection request $n$ moves back to $\mathcal{U}_k$.
24. 　　**end if**
25. 　**end for**
26. **end for**
27. **until** no connection request is willing to be a part of other groups.
28. **return** the sets of matching pairs.

---

### 3.2. Sub-Wavelength Grooming Algorithm

After obtaining the matching pairs, we can construct the sub-wavelength units which can be denoted by $\mathcal{U}^* = \{\mathcal{U}_1^*, \mathcal{U}_2^*, \cdots, \mathcal{U}_{\mathcal{I}}^*\}$, where $\mathcal{U}_i^*$ represents that a group of connection requests is aggregated into sub-wavelength $i$. The path set of the sub-wavelength units can be expressed as $L_{\mathcal{U}^*} = \left\{ L_{\mathcal{U}_1^*}, L_{\mathcal{U}_2^*}, \cdots, L_{\mathcal{U}_{\mathcal{I}}^*} \right\}$, where $L_{\mathcal{U}_i^*} = \max\{L_{\mathcal{S}_n}\}$, $\mathcal{S}_n \in \mathcal{U}_i^*$. To decrease the number of wavelengths, several sub-wavelengths are groomed into the same wavelength so long as the path affiliation $L_{\mathcal{U}_1^*} \subseteq L_{\mathcal{U}_2^*} \subseteq \cdots \subseteq L_{\mathcal{U}_k^*}$ is satisfied. We define the wavelength set as $\mathcal{W}$, and hence the wavelength-minimized problem can be transformed into

$$\min_{\mathcal{W}=\{\mathcal{W}_1, \mathcal{W}_2, \cdots, \mathcal{W}_\Phi\}} (\max|\mathcal{W}|) \tag{13}$$

$$s.t. \ \max|\mathcal{W}| = \Phi - Num(\mathcal{W}_\varphi)_{1 \leq \varphi \leq \Phi}^{\mathcal{W}_\varphi \cap \mathcal{U}^* = \varnothing} \tag{14}$$

$$L_{\mathcal{U}_1^*} \subseteq L_{\mathcal{U}_2^*} \subseteq \cdots \subseteq L_{\mathcal{U}_k^*} \tag{15}$$

where the $Num(\mathcal{W}_\varphi)_{1 \leq \varphi \leq \Phi}^{\mathcal{W}_\varphi \cap \mathcal{U}^* = \varnothing}$ is the number of idle wavelengths after the sub-wavelength grooming. Through the transformation, the binary variable $\mu_{ij}^w$ can be eliminated and the

analytical derivation is greatly simplified. In addition, the flow conservation limitation is converted into the path affiliation problem, as shown in (15). The SG algorithm is implemented by establishing a matching $\Re$ which serves a mapping between $\mathcal{U}^*$ and $\mathcal{W}$, so that for each sub-wavelength $\mathcal{U}_i^*$ and each wavelength $\mathcal{W}_\varphi$, we have: (i) $\Re(\mathcal{U}_i^*) = \mathcal{W}_\varphi$ if and only if $\mathcal{U}_i^* \in \Re(\mathcal{W}_\varphi)$; (ii) $\Re(\mathcal{W}_\varphi) \subseteq \mathcal{U}^* \cup \varnothing$ and $\left|\Re(\mathcal{W}_\varphi)\right| \leq \mathcal{I}$; (iii) $\Re(\mathcal{U}_i^*) \in \mathcal{W}$ and $\left|\Re(\mathcal{U}_i^*)\right| \leq 1$. Specifically, each matching pair is denoted by $\left(\mathcal{U}_i^*, \mathcal{W}_\varphi\right)_{L_{\mathcal{U}_1^*} \subseteq L_{\mathcal{U}_2^*} \subseteq \cdots \subseteq L_{\mathcal{U}_k^*}}$, satisfying the flow conservation constraints. We aim at finding a matching such that the consumed wavelengths can be minimized.

Given a matched sub-wavelength unit $(\mathcal{S}_n, \mathcal{U}_i)$, it prefers to be aggregated to form a new pair $(\mathcal{W}_{\varphi'}, (\mathcal{S}_n, \mathcal{U}_j))$ with more energy consumption and less used wavelengths. Therefore, we construct a preference matrix $Z_{n,j}$ to depict the mutual effect. Each element in $Z_{n,j}$ is defined as

$$\varsigma_{(\mathcal{W}_{\varphi'}, (\mathcal{S}_n, \mathcal{U}_j))}^{(\mathcal{S}_n, \mathcal{U}_i)} = \Xi(\mathcal{S}_n, \mathcal{U}_i)^{\rho_1 |\mathcal{W}_{\varphi'}|} / \Xi(\mathcal{S}_n, \mathcal{U}_j)^{\rho_2} \tag{16}$$

where $\rho_1$ and $\rho_2$ are the preference parameters. We then say that a sub-wavelength unit $(\mathcal{S}_n, \mathcal{U}_i)$ prefers $(\mathcal{W}_1, (\mathcal{S}_n, \mathcal{U}_1))$ to $(\mathcal{W}_2, (\mathcal{S}_n, \mathcal{U}_2))$ if $\varsigma_{(\mathcal{W}_1, (\mathcal{S}_n, \mathcal{U}_1))}^{(\mathcal{S}_n, \mathcal{U}_i)} > \varsigma_{(\mathcal{W}_2, (\mathcal{S}_n, \mathcal{U}_2))}^{(\mathcal{S}_n, \mathcal{U}_i)}$, i.e.,

$$(\mathcal{W}_1, (\mathcal{S}_n, \mathcal{U}_1)) \succ_{(\mathcal{S}_n, \mathcal{U}_i)} (\mathcal{W}_2, (\mathcal{S}_n, \mathcal{U}_2))$$
$$\Leftrightarrow \varsigma_{(\mathcal{W}_1, (\mathcal{S}_n, \mathcal{U}_1))}^{(\mathcal{S}_n, \mathcal{U}_i)} > \varsigma_{(\mathcal{W}_2, (\mathcal{S}_n, \mathcal{U}_2))}^{(\mathcal{S}_n, \mathcal{U}_i)} \tag{17}$$
$$\forall \mathcal{W}_1, \mathcal{W}_2 \in \mathcal{W}, \mathcal{U}_1 \neq \mathcal{U}_2$$

The attitude of each sub-wavelength unit $(\mathcal{S}_n, \mathcal{U}_i)$ towards the potential matching pair is influenced by the preference parameters $\rho_1$ and $\rho_2$. When $\rho_1 = 0$, the sub-wavelength unit tends to minimize the total energy consumption. When $\rho_2 = 0$, the unit only caters to minimize the number of wavelengths. Accordingly, $\rho_1 = \rho_2$ reflects a neutral state where the tradeoff between the total energy consumption and the number of used wavelengths can be achieved.

After obtaining the preference relation, a swap operation is implemented to optimize the matching structure. That is, a sub-wavelength unit tends to swap its matches with another sub-wavelength while keeping others sub-wavelength units unchanged. Specifically, for two existing matching pairs $(\mathcal{W}_1, (\mathcal{S}_n, \mathcal{U}_1))$ and $(\mathcal{W}_2, (\mathcal{S}_n, \mathcal{U}_2))$ which satisfy $\left|\varsigma_{(\mathcal{W}_1, (\mathcal{S}_n, \mathcal{U}_1))}^{(\mathcal{S}_n, \mathcal{U}_i)} + \varsigma_{(\mathcal{W}_2, (\mathcal{S}_n, \mathcal{U}_2))}^{(\mathcal{S}_n, \mathcal{U}_i)}\right| > \left|\varsigma_{(\mathcal{W}_1, (\mathcal{S}_n, \mathcal{U}_2))}^{(\mathcal{S}_n, \mathcal{U}_i)} + \varsigma_{(\mathcal{W}_2, (\mathcal{S}_n, \mathcal{U}_1))}^{(\mathcal{S}_n, \mathcal{U}_i)}\right|$, the swap operation is performed as

$$\Re_{(\mathcal{W}_1, (\mathcal{S}_n, \mathcal{U}_1))}^{(\mathcal{W}_2, (\mathcal{S}_n, \mathcal{U}_2))} = \Re \backslash \{(\mathcal{W}_1, (\mathcal{S}_n, \mathcal{U}_1)), (\mathcal{W}_2, (\mathcal{S}_n, \mathcal{U}_2))\}$$
$$\cup \{(\mathcal{W}_1, (\mathcal{S}_n, \mathcal{U}_2)), (\mathcal{W}_2, (\mathcal{S}_n, \mathcal{U}_1))\} \tag{18}$$

The whole sub-wavelength grooming algorithm is detailed in Algorithm 2. In the initialization step (line 2–5), each connection request is matched with a sub-wavelength. After that, the path affiliation is reconstructed, and the preference matrix is established. The following swap operation (line 7–19) targets finding a stable matching $\Re^*$. There are multiple iterations in each of which both the compare and update operations are performed. The iterations will not stop until the total number of wavelengths remains unchanged by the swap operation.

---

**Algorithm 2:** SG algorithm

---

**Input:** Sets of connection requests $\mathcal{S}$ and its corresponding paths $L_{\mathcal{S}}$, sub-wavelengths (groups) $\mathcal{U}$, and wavelength $\mathcal{W}$.
**Output:** A stable matching $\Re^*$.
1. **Initialization**
2. Perform **Algorithm 1** to obtain the optimal sub-wavelength unit $\mathcal{U}^*$.
3. Reconstruct the path affiliation $L_{\mathcal{U}_1^*} \subseteq L_{\mathcal{U}_2^*} \subseteq \cdots \subseteq L_{\mathcal{U}_k^*}$.
4. Calculate $\Xi(\mathcal{S},\mathcal{U})$ and construct the preference matrix $Z_{n,j}$.
5. Denote an initial sub-wavelength unit and wavelength as $\mathcal{U}_0^*$ and $\mathcal{W}_0$, respectively.
6. **Swap operation**
7. **Repeat**
8.   Select a matching pair $(\mathcal{W}_\varphi, (\mathcal{S}_n, \mathcal{U}^*)) \notin \Re$ satisfying the path affiliation.
9.   **if** $|\Re(\mathcal{W}_\varphi)| \leq \mathcal{I}$, and $\Re_{(\mathcal{W}_\varphi,(\mathcal{S}_n\mathcal{U}^*))}^{(\mathcal{W}_0,(\mathcal{S}_n\mathcal{U}_0^*))}$ is feasible, then
10.     Execute the swap matching and set $\Re = \Re_{(\mathcal{W}_\varphi,(\mathcal{S}_n\mathcal{U}^*))}^{(\mathcal{W}_0,(\mathcal{S}_n\mathcal{U}_0^*))}$.
11.     **else** keep looking for a feasible matching.
12.     Select two pairs $(\mathcal{W}_0, (\mathcal{S}_n, \mathcal{U}_0^*)) \in \Re$ and $(\mathcal{W}_1, (\mathcal{S}_n, \mathcal{U}_1^*)) \in \Re$
13.     **if** $|\varsigma_{(\mathcal{W}_1,(\mathcal{S}_n\mathcal{U}_0))}^{(\mathcal{S}_n\mathcal{U}_i)} + \varsigma_{(\mathcal{W}_0,(\mathcal{S}_n\mathcal{U}_1))}^{(\mathcal{S}_n\mathcal{U}_i)}| > |\varsigma_{(\mathcal{W}_1,(\mathcal{S}_n\mathcal{U}_1))}^{(\mathcal{S}_n\mathcal{U}_i)} + \varsigma_{(\mathcal{W}_0,(\mathcal{S}_n\mathcal{U}_0))}^{(\mathcal{S}_n\mathcal{U}_i)}|$, then
14.       Execute the swap matching and update $\Re = \Re_{(\mathcal{W}_1,(\mathcal{S}_n\mathcal{U}_0^*))}^{(\mathcal{W}_0,(\mathcal{S}_n\mathcal{U}_1^*))}$.
15.       **else** implement the swap matching and update $\Re = \Re_{(\mathcal{W}_1,(\mathcal{S}_n\mathcal{U}_1^*))}^{(\mathcal{W}_0,(\mathcal{S}_n\mathcal{U}_0^*))}$
16.     **end if**
17. **end if**
18. Update $\Xi(\mathcal{S},\mathcal{U})$, $Z_{n,j}$, and $\mathcal{W}$
19. **Until** the total number of wavelengths remains unchanged by the swap operation.
20. **Return** the stable matching $\Re^*$.

---

### 3.3. Property Analysis

The TPTG_MA is composed of the TAASA algorithm and the SG algorithm. In this subsection, the properties of the proposed TPTG_MA are discussed, including computational complexity, convergence, and stability.

(1) Computational complexity: For the TAASA algorithm, it is assumed that $N$ connection requests are aggregated into $M$ sub-wavelength units. In the phase of proposing and rejecting, each request will choose a sub-wavelength to match with. The computational complexity can be calculated as $\mathcal{O}(MN)$. Let $\omega$ be the number of iterations in each of which every connection request needs to visit $M-1$ sub-wavelength unit due to the compare-and-swap operations. Consequently, the total computational complexity of the TAASA algorithm can be expressed as $\mathcal{O}(MN + \omega M(N-1))$. For the SG algorithm, there are $NM\Phi$ combinations of matching pairs. In the initialization phase, we need to construct the sub-wavelength unit utilizing the TAASA algorithm. Assuming that the outcome of the SG algorithm remains unchanged after $\kappa$ iterations, the computational complexity of the SG algorithm can be calculated as $\mathcal{O}(\kappa NM\Phi(M\Phi - 1))$. Therefore, the computational complexity of the proposed TPTG_MA can be expressed as

$$
\begin{aligned}
&\mathcal{O}(MN + \omega M(N-1)) + \mathcal{O}(\kappa NM\Phi(M\Phi - 1)) \\
&= \mathcal{O}((\omega+1)MN + MN(\kappa M\Phi^2 - \kappa\Phi)) \\
&= \mathcal{O}(MN(\kappa M\Phi^2 + \omega - \kappa\Phi + 1))
\end{aligned} \tag{19}
$$

(2) Convergence and stability: In the TAASA algorithm, we allocate all connection requests to groups according to the path affiliation, which implies that the number of sub-wavelengths available for a request is limited. During the compare-and-swap phase, the moving of a single connection request from one group to another will influence the total energy consumption. When the minimum energy consumption is achieved, the connection request will not move anymore, and the matching structure remains stable. Therefore, the convergence and stability of the TAASA algorithm can be guaranteed. In the SG algorithm, the constraints, i.e., (i) $\Re(\mathcal{W}_\varphi) \subseteq \mathcal{U}^* \cup \varnothing$ and $|\Re(\mathcal{W}_\varphi)| \leq \mathcal{I}$; (ii) $\Re(\mathcal{U}_i^*) \in \mathcal{W}$ and

$\left|\Re(\mathcal{U}_i^*)\right| \leq 1$; (iii) the $\Re^{(\mathcal{W}_i,(\mathcal{S}_n,\mathcal{U}_i^*))}_{(\mathcal{W}_\varphi,(\mathcal{S}_n,\mathcal{U}^*))}$ is feasible, and are not violated after the swap operation. Hence, a stable matching will be obtained after several iterations.

## 4. Simulation and Performance Evaluation

In this section, the simulation results and analysis are presented to demonstrate the effectiveness of the proposed TPTG-MA. The simulation is conducted based on the MATLAB 2020 (a), where the topology generation and algorithm implementation are performed. The DSCNs are composed of DSCs with the star topology, and the DSCs communicate with each other through ISLs established between boundary nodes, as shown in Figure 2. The number in each link indicates the physical distance of the link in km, which is a critical parameter for routing selection. The test network topology is randomly generated, where different network sizes, such as 6 nodes, (e.g., DSC1), 12 nodes, (e.g., DSC1 and DSC4), and 22 nodes, can be obtained for further performance evaluation. It is assumed that the satellite beam switching does not happen during the simulation time. Therefore, the traffic grooming in this scenario is considered to be static.

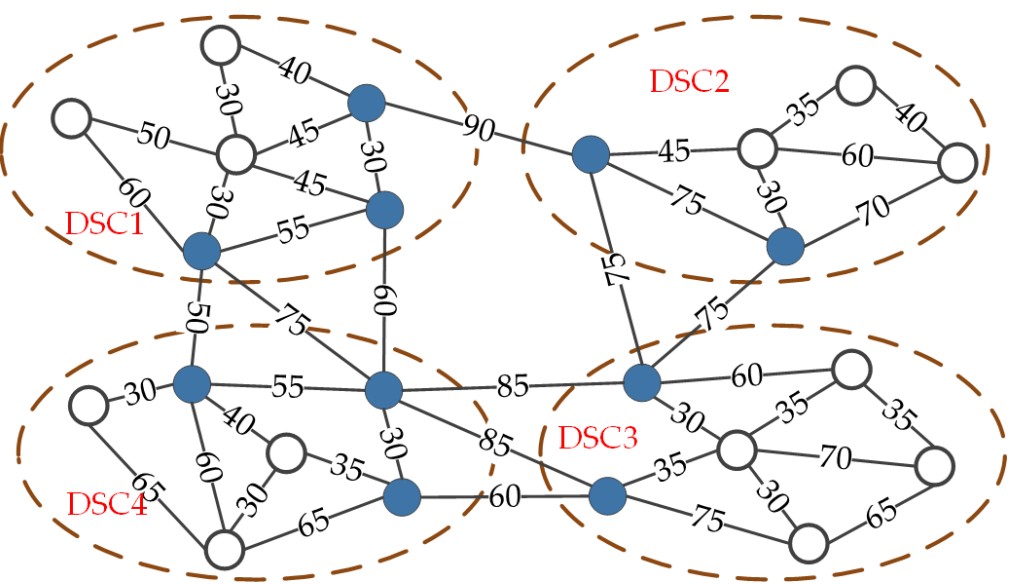

**Figure 2.** Test network topology in simulation.

There is no queuing mechanism, and once the capacity of the lightpath is insufficient, the traffic request will be blocked. The preference parameters $\rho_1$ and $\rho_2$ are both set to be 0.5. Other major simulation parameters are listed in Table 1. To evaluate the performance of the proposed TPTG_MA, the DLG heuristic and the GA [14,15] are simulated for comparison with respect to the AWUR, the ECS, the number of hops, and the blocking probability under the condition of different traffic intensity. Here, the blocking probability is the ratio of the number of traffic request failures to the number of total traffic requests, and the traffic intensity represents the product of the number of traffic requests per time unit and the duration of this simulation. Traffic requests are randomly generated within the bandwidth ranges. To determine the number of traffic requests between any node pair, we first construct a random matrix and make sure that the sum of all the elements in the matrix is 1. Then, the total number of traffic requests are distributed according to the weight of the matrix elements.

**Table 1.** Major simulation parameters.

| Parameters | Value |
|---|---|
| Bandwidth range of a traffic request | 20 Mbps–300 Mbps |
| Capacity of a sub-wavelength | 2 Gbps |
| Capacity of a wavelength | 10 Gbps |
| Ports for traffic aggregation | 40 |
| Ports for optical/electric conversion | 20 |
| Total optical/electric ports | 20/60 |
| Energy consumption per port $E_{oe}/E_{eo}/E_{agg}/E_{edfa}/E_{TX}$ | 15/15/5/10/20 W |

*4.1. Analysis of the AWUR*

The AWUR is an important parameter related to wavelength efficiency. Given the traffic requests and the used wavelengths, the AWUR can be calculated by

$$
\text{AWUR} = \frac{\sum\limits_{s,d \in V} \sum_{f \in [1,F_{k,d}]} b_{k,d}^f}{\Omega(\sum_w (\sum\limits_{i,j:i \neq j} \mu_{ij}^w + \sum\limits_{i',j':i' \neq j', P_{i'j'} \cap P_{ij} = \varnothing} \mu_{i'j'}^w))}
\tag{20}
$$

In Figure 3, the comparisons of the AWUR between the DLG, TPTG, DLG_GA, and TPTG_MA under different network sizes are presented. It can be observed that the AWUR of the TPTG_MA is obviously higher than that of other algorithms. In terms of heuristic, the proposed TPTG outperforms the DLG heuristic in improving the AWUR under different network sizes, which implies that more wavelength savings are achieved via the TPTG. For the six-node topology, the available links are limited. When the traffic intensity is beyond 300 Erl, the four algorithms achieve the same AWUR as shown in Figure 3a. This is because most of the wavelength capacity has been occupied and the traffic grooming algorithm has no effect on the improvement of AWUR.

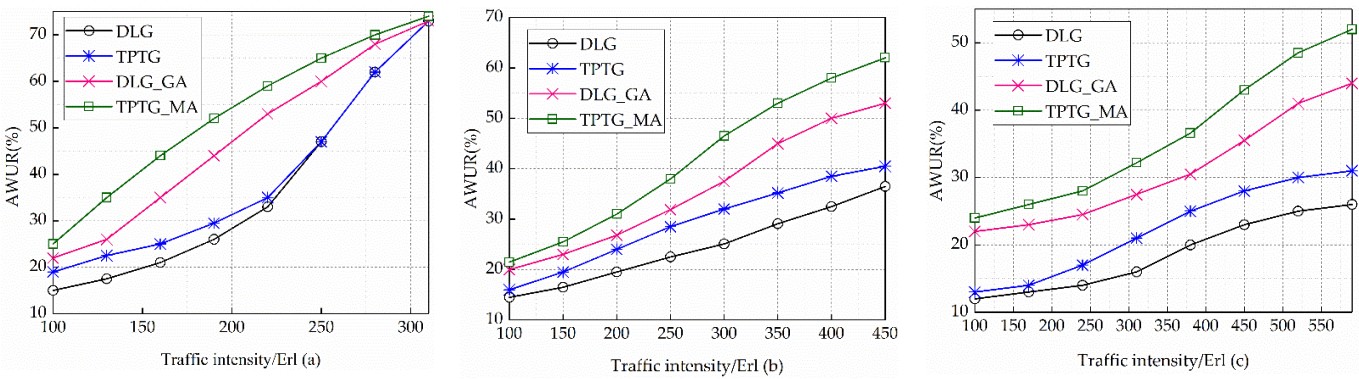

**Figure 3.** Comparisons of the AWUR between the DLG, TPTG, DLG_GA, and TPTG_MA under (**a**) 6-node topology, (**b**) 12-node topology, and (**c**) 22-node topology.

As the network size increases, we can observe a decrease in the AWUR under the same traffic intensity in Figure 3b,c. This is due to an increasing number of available wavelengths. Compared with the DLG_GA, the proposed TPTG_MA achieves higher AWUR, at most 18% improvement in Figure 3b and 23% improvement in Figure 3c. As the traffic intensity increases, Figure 3c witnesses a quick increase in the AWUR for the TPTG_MA, particularly when the traffic intensity is beyond 450 Erl. This is because the proposed TPTG_MA tries to groom as many traffic requests as possible into the existing wavelengths instead of occupying new wavelengths.

### 4.2. Analysis of the ECS

To evaluate the energy efficiency, we analyze the ECS which can be calculated by

$$\text{ECS} = \frac{E^{Total} - E^*_{total}}{E^{Total}} \qquad (21)$$

where $E^*_{total}$ represents the total energy consumption obtained by the traffic grooming algorithm. The ECS of the DLG, TPTG, DLG_GA, and TPTG_MA under different network sizes is shown in Figure 4. We can see that a significant ECS is achieved when the TPTG_MA and DLG_GA are implemented. This is because numerous traffic requests are aggregated into an identical wavelength leveraging the two algorithms, which decreases the unnecessary port consumption for optical/electric conversion. When the traffic intensity is 100 Erl, the TPTG_MA can achieve about 18%, 34%, and 40% ECS improvement over the TPTG for the 6-node, 12-node, and 22-node topology, respectively, which validates the effectiveness of the proposed MA in decreasing the energy consumption. As the traffic intensity increases, we can observe a decrease in the ECS for TPTG, DLG_GA, and TPTG_MA. Indeed, when the groomed traffic requests are closer to the maximum lightpath capacity, fewer grooming operations are performed, thereby leading to fewer ECS.

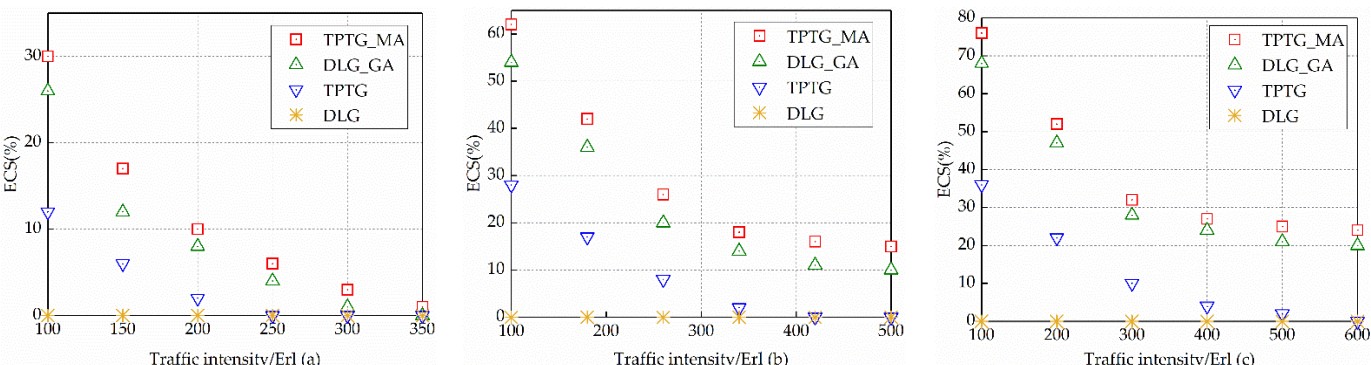

**Figure 4.** Comparisons of the ECS between the DLG, TPTG, DLG_GA, and TPTG_MA under (**a**) 6-node topology, (**b**) 12-node topology, and (**c**) 22-node topology.

In addition, the ECS highly depends on the network size. That is, as the network size increases, the ECS is greatly improved under the same traffic intensity. Taking the TPTG_MA for example, when the traffic intensity is 100 Erl, the 22-node topology can achieve about 32% and 47% ECS improvement over the 6-node and 12-node topology, respectively. Because there are many available wavelengths in the 22-node topology, where more grooming can be performed.

### 4.3. Analysis of the Number of Wavelengths and Hops

The following performance evaluations are based on the DSCNs with a 22-node topology. Figure 5 shows the comparison of the average number of wavelengths per node between the DLG, TPTG, DLG_GA, and TPTG_MA. As we can observe, the wavelength consumption per node of the TPTG_MA is fewer than that of the other three algorithms. When the traffic intensity is 300 Erl, the TPTG_MA consumes an average of 33.3%, 55.5%, and 66.7% few wavelengths per node than the DLG_GA, TPTG, and DLG, respectively. This is because the TPTG_MA grooms the traffic requests into the wavelengths in a finer way. In Section 2, we formulate the traffic grooming as a two-phase ILP problem, i.e., the first phase for TAASA, and the second phase for SG. In this way, most wavelengths are fully exploited, hence decreasing the number of wavelengths.

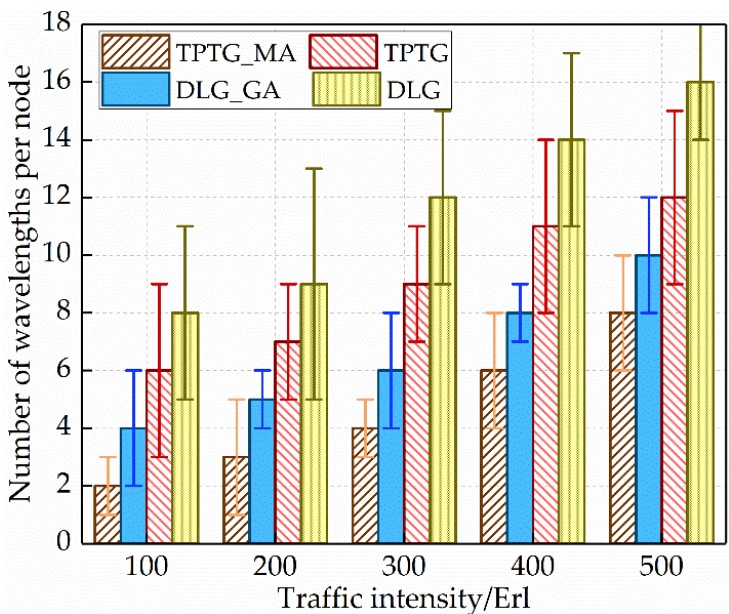

**Figure 5.** Comparison of the number of wavelengths per node between the DLG, TPTG, DLG_GA, and TPTG_MA.

Figure 6 presents the properties of the average number of hops per-flow between the proposed TPTG_MA and the other three algorithms. It can be observed that with the traffic intensity increasing, the average number of hops per-flow is improved. The average number of hops per-flow for the proposed TPTG_MA is slightly more than that of the other three algorithms. The reason is that the proposed TPTG_MA improves the utilization of wavelength capacity at the expense of elongating the hops of a few traffic requests. Although parts of traffic requests suffer the detour, the overall enhancement of network capacity utilization is more than the limited expense.

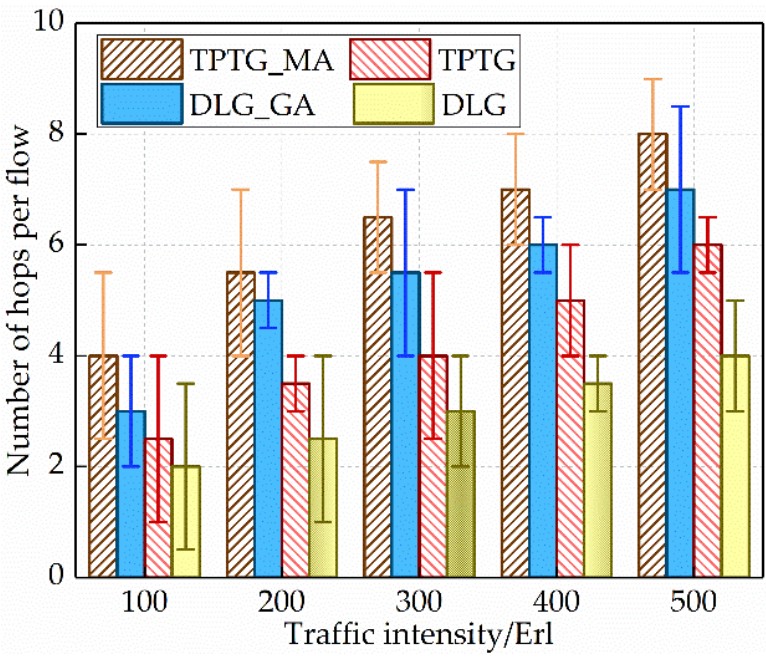

**Figure 6.** Comparison of the number of hops per-flow between the DLG, TPTG, DLG_GA, and TPTG_MA.

### 4.4. Analysis of Blocking Probability

As the average wavelength utilization ratio improves, the blocking probability of traffic requests cannot be ignored. Figure 7 presents the blocking probability between the proposed TPTG_MA and the other three algorithms. It can be observed that, with the utilization ratio of wavelength capacity increasing, the blocking probability significantly increases. This is because the wavelength capacity is insufficient to serve the increasing traffic requests. As shown in Figure 7, the blocking probability of the proposed TPTG_MA is lower than that of other algorithms. When the AWNR is 40%, the TPTG_MA can achieve 12%, 9%, and 3% blocking probability reduction compared to the DLG, TPTG, and DLG_GA, respectively. This is owing to the effective principle of flow matching between traffic requests, sub-wavelengths, and wavelengths.

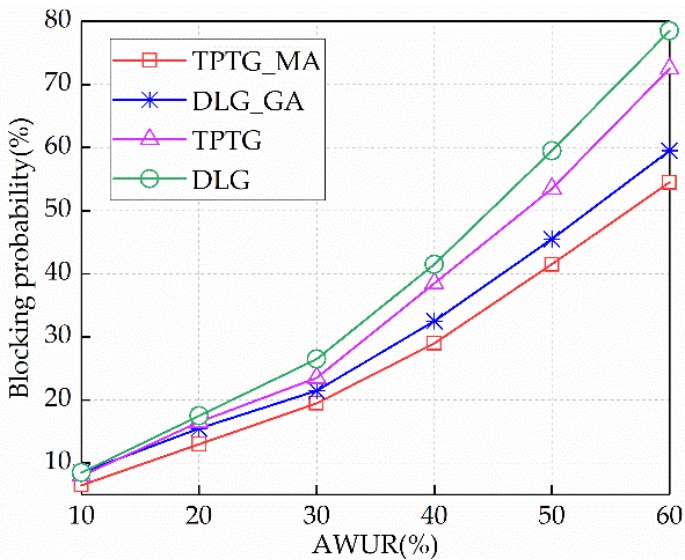

**Figure 7.** Comparison of the blocking probability between the DLG, TPTG, DLG_GA, and TPTG_MA.

### 4.5. Analysis of Convergence Property

Figure 8 is the comparison of convergence property between the TPTG_MA and DLG_GA. As can be seen, the convergence of the DLG_GA appears at the 50th iteration, while the proposed TPTG_MA begins to converge at about the 65th iteration. This is due to the massive compare-and-swap operations in the process of flow matching for the TPTG_MA.

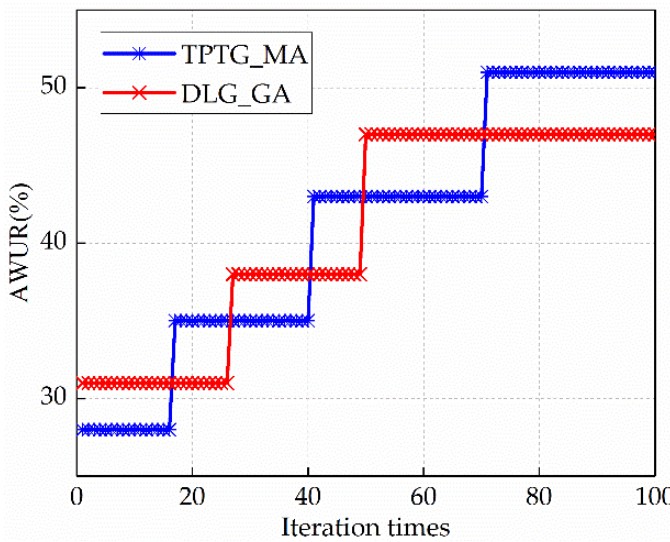

**Figure 8.** Comparison of convergence property between the TPTG_MA and DLG_GA.

## 5. Conclusions

In this paper, the energy- and wavelength-efficient traffic grooming based on the matching algorithm has been investigated in the DSCNs. Considering the limited wavelength and energy resource in the DSCNs, we, respectively, establish the energy- and wavelength- minimized models. To solve the problem, we introduce the sub-wavelength granularity and formulate the traffic grooming into a two-phase ILP problem, i.e., the first phase for TAASA and the second phase for SG. Then, we develop the TPTG_MA to resolve the ILP problem. Simulation results demonstrate that the TPTG_MA and DLG_GA outperform TPTG and DLG in the average wavelength utilization ratio, the energy consumption saving, and the blocking probability. Compared with the DLG_GA, the proposed TPTG_MA achieves at most 23% higher AWUR and 10% ECS improvement. When the AWNR is 40%, the TPTG_MA can achieve 12%, 9%, and 3% blocking probability reduction compared to the DLG, TPTG, and DLG_GA, respectively. However, the performance enhancement is at the expense of elongating the hops of a few traffic requests and increasing the number of iterations. This is worthwhile because the overall network capacity utilization is significantly improved. In the future, it may be interesting to investigate the placement of the hybrid cross-connect in the DSCNs when the traffic load changes.

**Author Contributions:** Conceptualization, C.P. and Y.H.; methodology, C.P.; software, C.P. and D.Y.; validation, H.F., Y.H. and S.Z.; formal analysis, C.P.; investigation, Y.H.; resources, Y.H.; data curation, C.P.; writing—original draft preparation, C.P.; writing—review and editing, S.Z.; visualization, D.Y.; supervision, S.Z.; project administration, H.F.; funding acquisition, Y.H. All authors have read and agreed to the published version of the manuscript.

**Funding:** This work was supported by National Key research and development plan of China (No. 2019YFB1803200).

**Institutional Review Board Statement:** Not applicable.

**Informed Consent Statement:** Not applicable.

**Data Availability Statement:** Not applicable.

**Conflicts of Interest:** The authors declare no conflict of interest.

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
