# Peer review of "Provision of Energy- and Wavelength-Efficient Traffic Grooming for Sparse WDM-Enabled Distributed Satellite Cluster Networks"

_photonics, doi:10.3390/photonics9070494_

Round 1
Reviewer 1 Report
The paper presents a technique to assign and shape the traffic to different wavelengths in a distributed satellite optical satellite network.
The problem is basically a high complexity optimization problem of resource allocation and routing. The system model and the proposed solution are not clear.
In particular:
1) In the system model, it is not clear the role of the final terrestrial terminals, since all the analysis is developed regardless of the fact that the traffic requests are actually related to a service where the If the system is actually composed just by the interconnection of optical satellite links, Fig. 1 should reflect it.
2) The model for the traffic requests is not considered. Are all the traffic requests uniformly distributed over the nodes?
3) The model for the sub-wavelength is not presented. What is the meaning of sub-wavelength in a WDM environment? If a global bandwidth is available for each wavelength, then several bandwidth requests can be allocated inside a wavelength up to the available bandwidth. On the other hand, if more wavelengths can be generated and managed, this simply increases the total number of available wavelengths. What is the meaning and the need of introducing a sub-wavelength?
4) A major concern comes from the fact that there is a big gap between the specification of the model and the analysis. The model is specified by a very large set of parameters (see Section 2.1.2), which are not considered in the analysis, in the algorithm and in the results. See for example the number of ports, the number of optical amplifiers, etc. Since the results consider just one value for some of the parameters, then it is useless to introduce such a large number of degrees of freedom which could change significantly the results, but are not considered.
5) Another major concern comes form the fact that optimization algorithms proposed to solve the allocation problem are not specified at all, it is said that "a connection request selects the sub-wavelength with the minimum sum energy consumption" (algorithm 1) or "execute a swap matching" (algorithm 2) again if a cost is lower, but this signifies in practice that an exhaustive search is performed, since there is no criterion for the limitation of the cases considered towards the minimum.
6) The so-called "traffic grooming" is not specified. Is is just a reallocation of some traffic requests to different sub-wavelengths? In any case see the comment above on the sub-wavelengths.
7) The principles of the other techniques used in the results as a benchmark or as comparison are not presented. Some comments on the techniques used by other methods to solve the same problem should be outlined.
Author Response
Manuscript ID: photonics-1732686 Type: research article
Title: Provision of Energy- and Wavelength-Efficient Traffic Grooming for Sparse WDM-Enabled Distributed Satellite Cluster Networks.
Dear Reviewer,
On behalf of my co-authors, I would like to thank you for your constructive comments and suggestions.
Your comments are very helpful to us in revising the manuscript, and we have carefully considered and responded to each suggestion. A red-lined version of the manuscript indicating where changes have been made is also included with our submission.
Thank you again for your consideration of our revised manuscript.
Sincerely,
Cong Peng
Air Force Engineering University, Shaanxi, China
Comment to the author
The paper presents a technique to assign and shape the traffic to different wavelengths in a distributed satellite optical satellite network.
The problem is basically a high complexity optimization problem of resource allocation and routing. The system model and the proposed solution are not clear.
- In the system model, it is not clear the role of the final terrestrial terminals, since all the analysis is developed regardless of the fact that the traffic requests are actually related to a service where the If the system is actually composed just by the interconnection of optical satellite links, Fig. 1 should reflect it.
Response:
We are grateful for your careful review and constructive suggestion. The role of the terrestrial terminals in Fig.1 is not so important, because our target is to investigate the traffic grooming in the distributed satellite cluster optical networks. Fig.1 just presents the diagrammatic sketch of remote transmission of the services via optical satellite networks. For instance, data flow 1, 2, and 3 originate from satellites 2, U, and 3, and then terminate at satellites 3, 2, and 4, respectively. To simplify the analysis, we neglect the impact of the specific terrestrial terminals and assume that satellites in the DSCNs serve as the source and destination nodes. It is equivalent to the result that satellites have collected the traffic requests of users within their coverage. We have pointed out it and marked the changes in red in the revised paper.
- The model for the traffic requests is not considered. Are all the traffic requests uniformly distributed over the nodes?
Response:
Thank you for pointing this out. We are sorry for the unclear traffic model. Because we roughly explained that “Traffic requests are randomly generated within the bandwidth and quantity ranges.”. The bandwidth range is easy to understand. In this paper, the traffic requests are not uniformly distributed over the nodes. We have explained it in detail in the revised paper, that is, “To determine the number of traffic requests between any node pair, we first construct a random matrix and make sure that the sum of all the elements in the matrix is 1. Then, the total number of traffic requests are distributed according to the weight of the matrix elements.” The changes are marked in red in the revised paper.
- The model for the sub-wavelength is not presented. What is the meaning of sub-wavelength in a WDM environment? If a global bandwidth is available for each wavelength, then several bandwidth requests can be allocated inside a wavelength up to the available bandwidth. On the other hand, if more wavelengths can be generated and managed, this simply increases the total number of available wavelengths. What are the meaning and the need of introducing a sub-wavelength?
Response:
We are grateful for your careful review and constructive questions. First, we will explain the meaning of sub-wavelength in a WDM environment. As a matter of fact, the sub-wavelength is a kind of intermediate granularity between the basic traffic requests and the wavelength, such as OC-1. By introducing the sub-wavelength granularity, we can achieve a further segmentation of the fixed-capacity wavelength, targeting to enhance the utilization efficiency of wavelengths. Direct lightpath grooming (DLG) refers to allocating several bandwidth requests directly inside a wavelength up to the available bandwidth, which causes the low wavelength capacity utilization.
Then, we will answer the second question “What are the meaning and the need for introducing a sub-wavelength?”. In fact, by introducing the subwavelength, the group transmission of traffic requests is realized, which helps to save transmission resources, especially for energy and wavelength channels. Considering the extremely limited wavelengths in the optical satellite networks, it is urgent to explore these efficient wavelength utilization methods.
- A major concern comes from the fact that there is a big gap between the specification of the model and the analysis. The model is specified by a very large set of parameters (see Section 2.1.2), which are not considered in the analysis, in the algorithm, and in the results. See for example the number of ports, the number of optical amplifiers, etc. Since the results consider just one value for some of the parameters, then it is useless to introduce such a large number of degrees of freedom which could change significantly the results but are not considered.
Response:
Thank you for pointing this out. As you mentioned, a very large set of parameters in this paper is utilized to describe the energy- and wavelength- minimized models. Many parameters don’t appear in the analysis, in the algorithm, and in the results, for example, the number of ports, the number of optical amplifiers, etc. However, these parameters are necessary conditions throughout the full paper. In this paper, most parameters are utilized to construct constraint conditions, such as C1~C7. Although our target is to minimize the energy consumption and used wavelengths, the process of calculating the objective function value is inseparable from these constraint conditions. For example, in “3.1 Traffic aggregation and sub-wavelength assignment algorithm”, the matching utility as Formula (9) is composed of a very large set of parameters and decision variables. The calculation of these parameters and variables must satisfy the constraints C1~C7, which have been pointed out and marked in red in the revised paper, In the “3.1 Sub-wavelength grooming algorithm”, we leverage the transformation to eliminate the binary variable, and hence the analytical derivation is greatly simplified. For the question “Since the results consider just one value for some of the parameters, then it is useless to introduce such a large number of degrees of freedom which could change significantly the results, but are not considered.” In section 4, the values for some of the parameters are given in Tab 1. Other parameter values can be calculated by the decision variables and given parameters. These parameters are included in constraints and objective functions, once the values of these parameters are determined, the objective functions are only influenced by the defined decision variables. Consequently, we can resort to heuristic algorithms to solve the energy- and wavelength- minimized traffic grooming problem.
- Another major concern comes from the fact that optimization algorithms proposed to solve the allocation problem are not specified at all, it is said that "a connection request selects the sub-wavelength with the minimum sum energy consumption" (algorithm 1) or "execute a swap matching" (algorithm 2) again if a cost is lower, but this signifies in practice that an exhaustive search is performed since there is no criterion for the limitation of the cases considered towards the minimum.
Response:
We are grateful for your careful review and constructive question. Firstly, we will explain the expression “a connection request selects the sub-wavelength with the minimum sum energy consumption” in algorithm 1. In the “3.1 Sub-wavelength grooming algorithm”, we calculate the total energy consumption in Formula (9) and take it as the matching utility. Leveraging the Formula (10) and Formula (11), the traffic requests and the sub-wavelength can be matched with each other to minimize the energy consumption. During the compare-and-swap phase, the moving of a single traffic request from one group to another will influence the total energy consumption. When the minimum energy consumption is achieved, the traffic request will not move anymore and the matching structure keeps stable. In “3.3 Property analysis”, we also confirm that the convergence and stability of the TAASA algorithm can be guaranteed. Secondly, we will explain the expression “execute a swap matching (algorithm 2) again if a cost is lower”. In “3.2 Sub-wavelength grooming algorithm”, the traffic requests, sub-wavelengths, and wavelengths can be treated as three sets of players to be matched with each other to minimize the energy consumption and the used wavelengths. The Formulas (13-15) are used for obtaining the minimum number of wavelengths. After that, we introduce a matching R and confirm that the matching structure is stable. To describe the total energy consumption for sub-wavelength grooming, we establish the preference relation as Formulas (16-18). In “3.3 Property analysis”, the convergence and stability of the SG algorithm are confirmed.
- The so-called "traffic grooming" is not specified. Is it just a reallocation of some traffic requests to different sub-wavelengths? In any case, see the comment above on the sub-wavelengths.
Response:
We are grateful for your careful review and constructive question. In this paper, the reallocation of some traffic requests to different sub-wavelengths is just one step of “traffic grooming”. The DSCNs face the challenges of explosively increasing traffic requests and the limited number of wavelengths. To satisfy the service transmission requirements and leverage the limited wavelength resources, we introduce the sub-wavelengths as the intermediate granularity, and then divide the traffic grooming into two phases, including the first phase for traffic aggregation and sub-wavelength assignment (TAASA) and the second phase for sub-wavelength grooming (SG). After introducing the sub-wavelength, the group transmission of services is realized, which helps to save transmission resources, especially for energy and wavelength channels.
- The principles of the other techniques used in the results as a benchmark or as a comparison are not presented. Some comments on the techniques used by other methods to solve the same problem should be outlined.
Response:
We are very grateful for your careful review. In the revised paper, we have rewritten it in the Abstract part, that is, “To evaluate the performance of the proposed TPTG_MA, the direct lightpath grooming (DLG) heuristic and the Genetic algorithm (GA) are simulated for comparison. The results demonstrate that the TPTG_MA and DLG_GA outperform TPTG and DLG in the average wavelength utilization ratio (AWUR), the energy consumption saving (ESC), and the blocking probability. Compared with the DLG_GA, the TPTG_MA achieves at most 18% and 23% higher AWUR in the 12-nodes and 22-nodes topologies, respectively. In addition, the TPTG_MA can actualize at most 10% ECS improvement over the DLG_GA.” The changes are marked in red.

Reviewer 2 Report
In this paper, the authors addressed the energy and wavelength efficient traffic grooming for sparse WDM-enabled distributed satellite cluster networks. The reliability has been evaluated. However, the paper needs major revision by addressing the following comments.
1. The distributed satellite cluster net-works (DSCNs) is adaptd to which scenario, such as LEO?
2. In abstract part, Quantitative results needs to be given.
3. In the Introduction part, authors should point out the current optical switching scheme of inter-satellite links.
4. In Section 3, the authors are suggested to provide the comparisons between the energy and wavelength efficient traffic grooming algorithm design in earth and inter-satellite links.
5. In the simulation part, the paper just illustrates that the configurations of Major simulation parameters without specifying the simulation software.
6. In the simulation part, the paper just illustrates the only qualitative analysis. The quantitative analysis is suggested to provide.
7. The overall quality of the English language and expressions needs to be improved.
ed satellite cluster net-
12
works (DSCNs)
Author Response
Manuscript ID: photonics-1732686 Type: research article
Title: Provision of Energy- and Wavelength-Efficient Traffic Grooming for Sparse WDM-Enabled Distributed Satellite Cluster Networks.
Dear Reviewer,
On behalf of my co-authors, I would like to thank you for your constructive comments and suggestions.
Your comments are very helpful to us in revising the manuscript, and we have carefully considered and responded to each suggestion. A red-lined version of the manuscript indicating where changes have been made is also included with our submission.
Thank you again for your consideration of our revised manuscript.
Sincerely,
Cong Peng
Air Force Engineering University, Shaanxi, China
Comment to the author
In this paper, the authors addressed the energy and wavelength efficient traffic grooming for sparse WDM-enabled distributed satellite cluster networks. The reliability has been evaluated. However, the paper needs major revision by addressing the following comments.
- The distributed satellite cluster networks (DSCNs) are adapted to which scenario, such as LEO?
Response:
Thank you for pointing this out. Compared with the GEO satellite, the access delay of the LEO or MEO satellite is obviously lower, whereas the access delay of the MEO and LEO satellite is less than 50ms and 10ms, respectively. Therefore, the distributed satellite cluster networks are more than suitable to provide service for users within its coverage. Each DSC is functionally equivalent to a huge LEO or an MEO satellite. In each DSC, satellites are relatively static with each other, thereby favoring establishing a steady optical topology. For different DSCs, the inter-satellite links are established between boundary nodes within each other's visual window.
- In the abstract part, Quantitative results need to be given.
Response:
We are grateful for your careful review and constructive suggestion. In the revised paper, we have rewritten the Abstract part and added the necessary quantitative results, that is, “To evaluate the performance of the proposed TPTG_MA, the direct lightpath grooming (DLG) heuristic and the Genetic algorithm (GA) are simulated for comparison. The results demonstrate that the TPTG_MA and DLG_GA outperform TPTG and DLG in the average wavelength utilization ratio (AWUR), the energy consumption saving (ESC), and the blocking probability. Compared with the DLG_GA, the TPTG_MA achieves at most 18% and 23% higher AWUR in the 12-nodes and 22-nodes topologies, respectively. In addition, the TPTG_MA can actualize at most 10% ECS improvement over the DLG_GA.” The changes are marked in red.
- In the Introduction part, the authors should point out the current optical switching scheme of inter-satellite links.
Response:
We are grateful for your careful review and constructive suggestion. There may be two kinds of switching schemes in each satellite node, including transparent optical switching and non-transparent photoelectric hybrid switching. Therefore, a hybrid switching structure is required to achieve the abovementioned functions. In the revised paper, we have explained the switching architecture in the introduction part, that is, “To actualize traffic grooming, the hybrid cross-connect architecture is maintained in every satellite node, including the optical cross-connect (OXC) unit which is functioned as wavelength routing operations, and the digital cross-connect (DXC) unit which is applied for electric switching and flow aggregation [16]”. The detailed switching structure can be found in Ref [16], where an optical switching unit and an electric switching unit are cascaded to achieve wavelength transparent forwarding and photoelectric hybrid switching.
- In Section 3, the authors are suggested to provide the comparisons between the energy and wavelength efficient traffic grooming algorithm design in earth and inter-satellite links.
Response:
Thank you for pointing this out. In this paper, our target is to implement energy- and wavelength-efficient traffic grooming for the distributed satellite cluster networks. Different from terrestrial optical networks, the DSCNs face the challenges of explosively increasing traffic requests, the limited number of wavelengths, and restricted energy provisioning. Therefore, we first establish the energy- and wavelength- minimized models and then resort to the matching algorithm to solve the traffic grooming problem. Here, the energy and wavelength efficient traffic grooming algorithm in this paper is also called the TPTG_MA. To simplify the analysis, we assume that satellites in the DSCNs serve as the source and destination nodes. It is equivalent to the result that satellites have collected the traffic requests of users within their coverage. Therefore, the traffic grooming is only performed in the inter-satellite links to minimize the wavelengths and energy consumption.
- In the simulation part, the paper just illustrates that the configurations of Major simulation parameters without specifying the simulation software.
Response:
We are grateful for your careful review and constructive suggestion. In the revised paper, we have specified the simulation software, that is, “The simulation is conducted based on the MATLAB 2020 (a), where the topology generation and algorithm implementation are performed.”
- In the simulation part, the paper just illustrates the only qualitative analysis. A quantitative analysis is suggested to provide.
Response:
We are grateful for your careful review and constructive suggestion. In the revised paper, we have provided the quantitative analysis in section 4. The detailed changes are as follows:
4.1 Analysis of AWUR
“For the 6-nodes topology, the available links are limited. When the traffic intensity is beyond 300Erl, the four algorithms achieve the same AWUR as shown in Fig. 3(a).”
“Compared with the DLG_GA, the proposed TPTG_MA achieves higher AWUR, at most 18% improvement in Fig. 3(b) and 23% improvement in Fig. 3(c). As the traffic intensity increases, Fig. 3(c) witnesses a quick increase of the AWUR for the TPTG_MA, particularly when the traffic intensity is beyond 450Erl.”
4.2 Analysis of the ECS
“When the traffic intensity is 100Erl, the TPTG_MA can achieve about 18%, 34%, and 40% ECS improvement over the TPTG for the 6-nodes, 12-nodes, and 22-nodes topology, respectively, which validates the effectiveness of proposed MA in decreasing the energy consumption.”
“Taking the TPTG_MA for example, when the traffic intensity is 100Erl, the 22-nodes topology can achieve about 32% and 47% ECS improvement over the 6-nodes and 12-nodes topology, respectively.”
4.3 Analysis of the number of wavelengths and hops
“When the traffic intensity is 300Erl, the TPTG_MA consumes an average of 33.3%, 55.5%, and 66.7% few wavelengths per node than the DLG_GA, TPTG, and DLG, respectively.”
4.4 Analysis of blocking probability
“When the AWNR is 40%, the TPTG_MA can achieve 12%, 9%, and 3% blocking probability reduction compared to the DLG, TPTG, and DLG_GA, respectively.”
4.5 Analysis of convergence property
“As can be seen, the convergence of the DLG_GA appears at the 50th iteration, while the proposed TPTG_MA begins to converge at about the 65th iteration.”
- The overall quality of the English language and expressions needs to be improved.
Response:
We are very grateful for your careful review. At the same time, we are sorry for the mistakes in this manuscript and the inconvenience they cause in your reading. The manuscript has been thoroughly revised, so we hope it can meet the journal’s standards.

Reviewer 3 Report
The paper addresses sparse wavelength division multiplexing applied in distributed satellite cluster networks and proposes a novel traffic grooming scheme that employs a sub-wavelength and a two-phase traffic grooming, namely traffic aggregation / sub-wavelength assignment followed by sub-wavelength grooming.
The paper provides sufficient background information and the relevant literature, a scientifically sound theoretical analysis (section 2) a full description of the designed algorithm (section 3) and a fair set of simulation results (section 4).
My only comment regards the “Conclusions” section which I find rather short. I think it would be to the benefit of the paper this to be enhanced and, for example, include an evaluative overview of the obtained results rather than merely a review of the work done.
The paper is well written and, regarding the use of English, only a minor revision is needed.
Author Response
Manuscript ID: photonics-1732686 Type: research article
Title: Provision of Energy- and Wavelength-Efficient Traffic Grooming for Sparse WDM-Enabled Distributed Satellite Cluster Networks.
Dear Reviewer,
On behalf of my co-authors, I would like to thank you for your constructive comments and suggestions.
Your comments are very helpful to us in revising the manuscript, and we have carefully considered and responded to each suggestion. A red-lined version of the manuscript indicating where changes have been made is also included with our submission.
Thank you again for your consideration of our revised manuscript.
Sincerely,
Cong Peng
Air Force Engineering University, shaanxi, China
Comment to the author
The paper addresses sparse wavelength division multiplexing applied in distributed satellite cluster networks and proposes a novel traffic grooming scheme that employs a sub-wavelength and a two-phase traffic grooming, namely traffic aggregation / sub-wavelength assignment followed by sub-wavelength grooming.
The paper provides sufficient background information and the relevant literature, a scientifically sound theoretical analysis (section 2) a full description of the designed algorithm (section 3) and a fair set of simulation results (section 4).
My only comment regards the “Conclusions” section which I find rather short. I think it would be to the benefit of the paper this to be enhanced and, for example, include an evaluative overview of the obtained results rather than merely a review of the work done.
Response:
We are grateful for your careful review and constructive suggestion. We have improved the “Conclusions” section according to your suggestion, that is, “Simulation results demonstrate that the TPTG_MA and DLG_GA outperform TPTG and DLG in the average wavelength utilization ratio, the energy consumption saving, and the blocking probability. Compared with the DLG_GA, the proposed TPTG_MA achieves at most 23% higher AWUR and 10% ECS improvement. When the AWNR is 40%, the TPTG_MA can achieve 12%, 9%, and 3% blocking probability reduction compared to the DLG, TPTG, and DLG_GA, respectively.”

Round 2
Reviewer 1 Report
The authors has provided an answer to most of the requirements of the reviewers.However most of the answers are provided to the reviewer and not explained in the paper. The request is motivated by the need to increase readability to a general public of readers of Photonics.
Thus, I suggest to explain better also in the paper the answers given to the reviewer in particular points 3, 4, 5, 6 of the previous review:
- the idea of sub-wavelength and how this is different from an increased number of wavelengths
- the gap between the huge list of parameters of Table 1 and the expressions in the paper which do not contain any of them,
- the specification of the optimization algorithms
Author Response
Manuscript ID: photonics-1732686 Type: research article
Title: Provision of Energy- and Wavelength-Efficient Traffic Grooming for Sparse WDM-Enabled Distributed Satellite Cluster Networks.
Dear Reviewer,
On behalf of my co-authors, I would like to thank you for your constructive comments and suggestions.
Your comments are very helpful to us in revising the manuscript. In order to increase readability to general public of readers of Photonics, we have carefully considered and responded to each suggestion. The changes are marked in red and underlined in the revised paper.
Thank you again for your consideration of our revised manuscript.
Sincerely,
Cong Peng
Air Force Engineering University, shaanxi, China
Comment to the author
The authors have provided an answer to most of the requirements of the reviewers. However, most of the answers are provided to the reviewer and not explained in the paper. The request is motivated by the need to increase readability to a general public of readers of Photonics.
Thus, I suggest to explain better also in the paper the answers given to the reviewer in particular points 3, 4, 5, 6 of the previous review:
1.the idea of sub-wavelength and how this is different from an increased number of wavelengths
Response:
We are grateful for your careful review and constructive suggestion. We have explained it in the revised paper: "On the one hand, we can achieve a further segmentation of the fixed-capacity wavelength, targeting to enhance the utilization efficiency of wavelengths. On the other hand, the group transmission of traffic requests is realized, which helps to save transmission resources especially for energy and wavelength channels.” The changes are marked in red and underlined in Section 1.
2. the gap between the huge list of parameters in Table 1 and the expressions in the paper which do not contain any of them
Response:
We are grateful for your careful review and constructive suggestion. We have explained it in the revised paper, that is, “The parameters defined in the last section are included in constraints and objective functions, once the values of these parameters are determined, the objective functions are only influenced by the defined decision variables. Consequently, we can resort to heuristic algorithms to solve the energy- and wavelength- minimized traffic grooming problem.” The changes are marked in red and underlined in Section 3.
3. the specification of the optimization algorithms
Response:
We are grateful for your careful review and constructive question. In the “3.1 Sub-wavelength grooming algorithm”, we calculate the total energy consumption in Formula (9) and take it as the matching utility. Leveraging Formula (10) and Formula (11), the traffic requests and the sub-wavelength can be matched with each other to minimize energy consumption. During the compare-and-swap phase, the moving of a single traffic request from one group to another will influence the total energy consumption. When the minimum energy consumption is achieved, the traffic request will not move anymore and the matching structure keeps stable. In “3.3 Property analysis”, we also confirm that the convergence and stability of the TAASA algorithm can be guaranteed. Secondly, we will explain the expression “execute a swap matching (algorithm 2) again if a cost is lower”. In “3.2 Sub-wavelength grooming algorithm”, the traffic requests, sub-wavelengths, and wavelengths can be treated as three sets of players to be matched with each other to minimize the energy consumption and the used wavelengths. The Formulas (13-15) are used for obtaining the minimum number of wavelengths. After that, we introduce a matching R and confirm that the matching structure is stable. To describe the total energy consumption for sub-wavelength grooming, we establish the preference relation as Formulas (16-18). In “3.3 Property analysis”, the convergence and stability of the SG algorithm are confirmed. Related explanations are underlined in Section 3.

Reviewer 2 Report
This paper can be accepted
Author Response
Manuscript ID: photonics-1732686 Type: research article
Title: Provision of Energy- and Wavelength-Efficient Traffic Grooming for Sparse WDM-Enabled Distributed Satellite Cluster Networks.
Dear Reviewer,
On behalf of my co-authors, I would like to thank you for your approval of this paper.
Thank you again for your consideration of our manuscript.
Sincerely,
Cong Peng
Air Force Engineering University, Shaanxi, China